# Immunogenicity and safety of the MF59-adjuvanted seasonal influenza vaccine in non-elderly adults: A systematic review and meta-analysis

**Alexander Domnich**[1], **Carlo-Simone Trombetta**[2], **Elettra Fallani**[3], **Marco Salvatore**[3,4]*

**1** Hygiene Unit, San Martino Policlinico Hospital—IRCCS for Oncology and Neurosciences, Genoa, Italy, **2** Department of Health Sciences (DISSAL), University of Genoa, Genoa, Italy, **3** Seqirus S.r.l., Monteriggioni (Siena), Italy, **4** Department of Life Sciences; University of Siena, Siena, Italy

* marco.salvatore@seqirus.com

**Data Availability Statement:** All raw data are within the paper and associated supporting materials.

## Abstract

### Objective

In Europe, the age indication for the MF59-adjuvanted quadrivalent influenza vaccine (aQIV) has recently been extended from $\geq$65 to $\geq$50 years. Considering that the earliest approval of its trivalent formulation (aTIV) in Italy was for people aged $\geq$12 years, we aimed to systematically appraise data on the immunogenicity, efficacy, and safety of aTIV/aQIV in non-elderly adults.

### Methods

A systematic literature review was conducted according to the available guidelines and studies were searched in MEDLINE, Biological Abstracts, Web of Science, Cochrane Library and clinical trial registries. Studies on absolute and relative immunogenicity, efficacy, effectiveness, and safety of aTIV/aQIV in non-elderly adults (<65 years) were potentially eligible. These endpoints were analyzed by virus (sub)types and characteristics of vaccinees. Fixed- and random-effects meta-analyses were performed for data synthesis. Protocol registration: CRD42024512472.

### Results

Twenty-four publications were analyzed. aTIV/aQIV was more immunogenic than non-adjuvanted vaccines towards vaccine-like strains: the absolute differences in seroconversion rates were 8.8% (95% CI: 3.7%, 14.0%), 13.1% (95% CI: 6.7%, 19.6%) and 11.7% (95% CI: 7.2%, 16.2%) for A(H1N1), A(H3N2), and B strains, respectively. This immunogenicity advantage was more pronounced in immunosuppressed adults. Additionally, aTIV/aQIV was more immunogenic than non-adjuvanted counterparts towards heterologous A(H3N2) strains with a 10.7% (95% CI: 3.2%, 18.2%) difference in seroconversion rates. Data on antibody persistence and efficacy were limited and inconclusive. Overall, aTIV/aQIV was

**Funding:** This work was supported by Seqirus S.r. L. that provided AD and CST with a fee. Seqirus S.r. L. was involved in study design, decision to publish, and preparation of the manuscript.

**Competing interests:** EF and MS are employees Seqirus S.r.L. AD was previously an employee of Seqirus S.r.L. AD and CST received a fee from Seqirus S.r.L. for conducting this systematic review. This does not alter our adherence to PLOS ONE policies on sharing data and materials. There are no patents, products in development or marketed products associated with this research to declare.

judged safe and well tolerated, although reactogenic events were more frequent in aTIV/ aQIV recipients versus comparators. Serious adverse events were uncommon and no difference (risk ratio 1.02; 95% CI: 0.64, 1.63) between aTIV/aQIV and non-adjuvanted formulations was found.

## Conclusions

In non-elderly adults, aTIV/aQIV is safe and generally more immunogenic than non-adjuvanted standard-dose vaccines.

## 1. Introduction

Trivalent (aTIV) and quadrivalent (aQIV) MF59-adjuvanted seasonal influenza vaccines have been developed to enhance the immune response to vaccination and potentially improve vaccine effectiveness. Until recently, aTIV and aQIV were licensed for adults aged ≥65 years. In the last 20 years, multiple studies have demonstrated the advantages of aTIV/aQIV over non-adjuvanted standard-dose formulations [1,2]. For instance, two systematic reviews of studies in older adults have shown that the use of aTIV induces higher immune response towards homologous vaccine-like [3,4], and heterologous influenza virus strains [4], and generally improved effectiveness against several influenza-related outcomes [5,6]. These advantages are attributable to the MF59 adjuvant, which exercises its immunostimulatory effect in several ways. Briefly, MF59 allows the recruitment of key immune cells to the injection site to make antigen uptake and transport to the lymph nodes more efficient. It then promotes T-cell activation and improves B-cell expansion, which leads to a greater number and diversity of antibodies [7,8].

Historically, most of the research on the burden of influenza focused on older adults, since they are the primary target group for annual vaccination [9]. Indeed, most severe complications, hospitalizations, and deaths related to influenza occur in the elderly population [10,11]. However, influenza attack rates are typically inversely related to age. The cumulative incidence of influenza in non-elderly adults is estimated to be about twice that of older adults (8.9% vs. 3.9%) in the United States (US) [12]. Younger adults comprise most of the workforce, and thus play a crucial role in socioeconomic welfare [13]. Up to 88% of the economic burden of influenza in non-elderly adults is attributable to indirect costs, mostly related to absenteeism and loss of productivity [9]. Nonetheless, the efficacy of the available influenza vaccines in non-elderly adults is suboptimal: a Cochrane meta-analysis estimated the efficacy of inactivated vaccines in healthy adults to be 59% [14].

In November 2023, the European Committee for Medicinal Products for Human Use (CHMP) adopted a positive opinion on the extension of the age indication for aQIV from ≥65 to ≥50 years [15]. Many may be unaware that the age indication for aTIV was ≥12 years when it was first approved in Italy in 1997. It was particularly indicated for older adults who are subject to immunosenescence, and individuals with immunosuppressive conditions [16]. In the subsequent few years, this age indication was changed to ≥65 years.

In Italy, aQIV and high-dose non-adjuvanted vaccines are preferentially recommended to older adults aged ≥65 years [17]. This is in line with the recommendations in other countries, such as the United Kingdom [18], US [19], and Australia [20]. As of February 2024, no specific recommendations on the use of aQIV in adults aged 50–64 years have been made in Italy or other European countries. Given the historical availability of aTIV in Italy for people aged ≥12

years [16], we anticipated that studies of aTIV in younger adults had been conducted. The objective of this review was to systematically collect and evaluate available experimental and observational data on the immunogenicity, efficacy, effectiveness and safety of aTIV/aQIV in non-elderly adults.

## 2. Methods

### 2.1. Review protocol and reporting standards

A protocol for this review was prospectively registered with the international, prospective register of systematic reviews (PROSPERO; ID: CRD42024512472) and no amendments were made afterwards. This review conforms to the preferred reporting items for systematic reviews and meta-analyses (PRISMA) statement [21] (S1 Table).

### 2.2. Eligibility criteria

The eligibility criteria were formulated using the PICO (population, intervention, comparison, outcome) framework [22]. The population of interest was working-age adults aged <65 years regardless of the presence of underlying health conditions and other characteristics. The intervention consisted of a single dose of the authorized formulations aTIV or aQIV. Formulations of the historically available aTIV and currently available aQIV are almost identical except for the inclusion of an additional B strain in aQIV. According to the aQIV summary of product characteristics [23], data on aTIV are also relevant for aQIV because both vaccines are manufactured using the same process, and have overlapping compositions. Furthermore, following the likely extinction of the B/Yamagata lineage during the Coronavirus disease 2019 (COVID-19) pandemic, the World Health Organization (WHO) has recently recommended the removal of the B/Yamagata component, and thus a return to trivalent vaccine formulations [24]. For these reasons, we treated aTIV and aQIV interchangeably. For the comparison, both non-active (placebo, non-influenza vaccines or non-vaccination) and active (any type of non-adjuvanted trivalent [TIV] or quadrivalent [QIV] influenza vaccines) comparators were considered. The study outcomes included several endpoints related to the domains of immunogenicity, efficacy, reactogenicity, and safety (detailed in section 2.3). We planned to assess effectiveness of aTIV/aQIV, but no studies were identified. For all outcomes, experimental and observational studies of any design were eligible.

The following were set as exclusion criteria: (i) reviews, modeling studies, and other secondary publications without original data; (ii) non-authorized experimental formulations of aTIV/aQIV (e.g. different amount of the antigen or MF59); (iii) pandemic monovalent formulations of the MF59-adjuvanted vaccines; (iv) vaccines adjuvanted with non-MF59 adjuvants (e.g. AS03, virosomes); (v) mixed study population of elderly and non-elderly adults with no separate data on the latter. With regards to the last criterion, we included studies if the majority of participants were <65 years.

### 2.3. Study outcomes

Our primary endpoints were humoral immunogenicity outcomes including different statistical parameters related to antibody titers measured in the hemagglutination-inhibition (HAI) assay. The HAI titer ≥1:40 is a universally recognized correlate of protection [25]. For the absolute humoral immune response of aTIV/aQIV (i.e. with no respect to comparators), seroconversion rates (SCRs) and seroprotection rates (SPRs) were primary endpoints. SCR was defined as the proportion of vaccinees with at least four-fold increase in HAI titers from before to after vaccination, while SPR was defined as the proportion of vaccinees reaching the HAI

titer ≥1:40 post-vaccination [25,26]. According to the Center for Biologics Evaluation and Research (CBER) criteria for the accelerated approval of inactivated influenza vaccines, the lower bounds of the two-sided 95% confidence intervals (CIs) of SCRs and SPRs in adults aged <65 years should be at least 40% and 70%, respectively [27]. Additionally, as some studies suggested that the HAI titer ≥1:40 may be insufficient [26,28] and some early studies on aTIV in older adults used a more conservative HAI threshold of ≥1:160 [29], this latter definition of SPR was also considered and tested in a sensitivity analysis.

For the relative humoral immune response (i.e. compared with non-adjuvanted vaccines), our primary endpoints were differences in SCRs (defined as $SCR_{aTIV/aQIV} - SCR_{Comparator}$), SPRs ($SPR_{aTIV/aQIV} - SPR_{Comparator}$), and post-vaccination geometric mean titer (GMT) ratios (GMTRs; defined as $GMT_{aTIV/aQIV} : GMT_{Comparator}$).

All serological parameters were analyzed by influenza vaccine-like strains, including A (H1N1), A(H3N2), and B for trivalent formulations, and A(H1N1), A(H3N2), and B/Victoria and B/Yamagata for quadrivalent formulations. Immune response towards heterologous or drifted strains was also considered. Serological parameters measured at approximately 1 month post-vaccination were of primary interest. Secondary endpoints were immunogenicity assessments performed at later time periods (up to 12 months post-vaccination), which may indicate the duration of vaccine-induced protection [30].

Additional endpoints included humoral immunogenicity measured in other serological tests (e.g. neutralization, single radial hemolysis assays, enzyme-linked immunosorbent assay [ELISA]) and cell-mediated immune response measured with any technique [25].

Efficacy was defined as reduction in the risk of influenza in individuals immunized with aTIV/aQIV, compared with placebo/non-influenza vaccines (absolute efficacy) and any available non-adjuvanted vaccine (relative efficacy), as estimated from randomized controlled trials (RCTs) [31].

For evaluation of reactogenicity and safety, we considered the frequency of both solicited (i.e. actively collected, pre-specified list of events, usually collected through diaries within the first week post-vaccination) and unsolicited (any events other than solicited events that are typically collected for the entire study duration) adverse events (AEs) recorded through RCTs. For injection site-solicited reactions, any local event, pain, erythema, induration, and ecchymosis were pre-specified. For systemic events, any systemic event, fever (≥38°C), chills, myalgia, arthralgia, headache, malaise, nausea, and rash were of interest. Within unsolicited events, frequency of serious AEs (SAEs) and SAEs judged to be vaccine-related were considered. SAEs were defined as events that resulted in death, caused persistent or significant disability or incapacity, were life-threatening or required hospitalization [32].

## 2.4. Search strategy

The search was first conducted in MEDLINE (via Ovid), Biological Abstracts (via Ovid), Web of Science and Cochrane Library. In all searches, no restrictions (e.g. language or year of publication) were applied. Additionally, the ClinicalTrials.gov prospective registry was searched for completed studies. The last automatic search was performed on 16 February 2024. We used a combination of medical subject headings (MeSH) and text-wide terms. The database-specific search scripts are reported in S2 Table.

A manual search was then performed. First, a standard backward cross-reference check of the included studies was conducted. We then performed a forward citation search by using Google Scholar (https://scholar.google.com/). Finally, vaccine manufacturers were asked to suggest other relevant studies; in particular those that are not yet published and/or have only been presented at conferences.

## 2.5. Study selection

Search outputs from each citation database were pooled in a single spreadsheet and duplicates were removed. Titles and abstracts were then screened against the inclusion and exclusion criteria, and clearly irrelevant records were excluded. Full texts of the remaining records were located and assessed for their eligibility. The list of studies identified through the automatic search was eventually finalized by adding those located through the manual search. Study selection was performed by two authors, AD and CST, each working independently. Eventual conflicts were solved by consensus.

## 2.6. Data extraction and abstraction

Relevant data were extracted from the full text and/or associated supplementary materials into an *ad hoc* spreadsheet. The following data items were extracted: (i) citation record; (ii) study location; (iii) influenza season; (iv) study design; (v) study population and main population characteristics (age and co-morbidities); (vi) sample size per study arm; (vii) outcome domains evaluated in the study; (viii) vaccine and test strains used; (ix) point estimates for the outcomes of interest with any available dispersion measures; (x) funding source; (xii) other potentially relevant information.

Data of interest that were not readily extractable from the primary publication record were handled as follows. The main publication source was firstly cross-checked against other sources (e.g. results posted on the clinical trial registry). For dichotomous outcomes (SCR, SPR, frequency of AEs) missing numerators or denominators were imputed arithmetically from the available percentages. Results presented only in graphical-form data were imputed using the WebPlotDigitizer v.4.6 online web application (https://automeris.io/WebPlotDigitizer).

Data extraction was performed by AD and cross-checked by CST.

## 2.7. Risk of bias

The risk of bias (RoB) in randomized studies was appraised by using the revised Cochrane RoB 2 tool [33]. For non-randomized trails, the RoB in non-randomized studies of interventions (ROBINS-I) tool [34] was used. The RoB tools were applied separately by AD and CST and disagreements were solved by consensus.

## 2.8. Data synthesis

A qualitative synthesis was provided first by visualizing tabulated data and forest plots. For the quantitative synthesis of single proportions related to the absolute immunogenicity and safety parameters of aTIV/aQIV, a proportional meta-analysis with double arcsine transformation to stabilize variances [35] was performed. For binary outcomes of the relative immunogenicity and safety parameters of aTIV/aQIV vs. non-adjuvanted vaccines, a binary meta-analysis using the Mantel-Haenszel's method was performed. The summary effect measures with 95% Clopper-Pearson exact CIs were risk difference for SCRs (ΔSCR) and SPRs (ΔSPR) [27] and risk ratio (RR) for AEs [36]. We also planned *a priori* to conduct a pooled analysis on log-transformed GMTs for aTIV/aQIV vs. TIV/QIV, as well as absolute and relative aTIV/aQIV efficacy/effectiveness.

In all pooled analyses, both fixed-effects (FE) and random-effects (RE) models were applied. RE models are an appropriate choice, as it captures heterogeneity between studies [37], which is expected to be high especially for the proportional meta-analysis [38]. However, when the number of included studies (k) is small, the estimated heterogeneity may be biased [39] and a

FE model may be considered [37]. Heterogeneity was measured by means of $\tau^2$ and $I^2$ statistics. In all analyses, the 95% prediction intervals (PIs) were also computed.

A subgroup analysis by the presence of immunosuppressive conditions was conducted. A univariable meta-regression analysis was conducted to explore the influence of study characteristics on the outcomes of interest. This analysis was performed only for analyses with k ≥10 [40].

Publication bias was assessed by visualizing funnel plots. This was evaluated only for meta-analyses with k ≥10 [40].

We performed three sensitivity analyses. The first concerned the operational definitions of SPR, which were described in section 2.3. In the second, we excluded studies with overlapping study populations (i.e. studies that also enrolled elderly participants). Thirdly, the trim-and-fill procedure [41] was implemented to estimate the number of potentially unavailable studies and to adjust pooled estimates for publication bias.

Quantitative synthesis was performed in R environment (R Foundation for Statistical Computing; Vienna, Austria) using the packages "meta" v. 7.0–0 and "metaphor" v. 4.4–0.

## 3. Results

### 3.1. Characteristics of the studies included

The automatic literature search resulted in 2,880 records. Following de-duplication, 2,064 titles and abstracts were screened, and 34 records were judged potentially eligible. Of 34 full texts evaluated, 22 [42–63] met the inclusion criteria and were retained. Twelve publications were excluded with reasons (S3 Table). Two additional citations [64,65] were found by manual search. The twenty-four records that were included corresponded to 18 vaccination cohorts. RCTs by Baldo et al. [50] and Kumar et al. [54] had extensions [51,55], in which sera were reanalyzed for the purpose of cross-reactive immune response. Similarly, a research group reported results of HAI testing towards different homologous and heterologous strains in two publications [47,64]. Fenoglio et al. [49] reported additional results on a subset of participants from an RCT report by Durando et al. [48], and the results of the exploratory BIOVAXSAFE trial were published in three different records [56–58]. A flowchart of the study selection process is depicted in Fig 1.

The main characteristics of the included studies are summarized in Table 1. Briefly, the available studies were conducted between 1995/1996 and 2020/2021 northern hemisphere influenza seasons, and 50% (9 of 18) came from Italy [43,44,46–48,50,52,53,65]. Sample size was in the range of 17–2,044 participants (median 98); cumulatively the trials enrolled 4,628 participants, of which 50.2% received one dose of aTIV/aQIV. Most (72%; 13/18) studies [42,43,45,46,48,50,54,56,59,60,62,63,65] were randomized and controlled in design. A trial by Iorio et al. [44] was controlled, but the allocation was systematic, while the remaining four studies [47,52,53,61] reported results of single-arm trials. The study population of eight studies [42,47,50,52,56,61,63,65] was composed of immunocompetent adults aged <65 years with or without co-morbidities. Conversely, the remaining 10 studies evaluated the immunogenicity, efficacy, and/or safety of aTIV in immunocompromised cohorts, including HIV-seropositive individuals [44,46,48,53], solid organ [43,45,54,62] and hematopoietic stem [60] cell transplant recipients, and those with end-stage renal disease [59]. Although the majority of study populations were composed of working-age adults, four of the studies on transplant recipients [43,54,60,62] also included elderly individuals. The trivalent formulation was used in all studies [42–62,64,65] except one [63], which used aQIV. Finally, nine trials were publicly funded [46,47,50,52,53,54,59,60,62], and eight studies were sponsored by the vaccine manufacturer [42,44,45,48,56,61,63,65], while the funding source was not disclosed in one study [43].

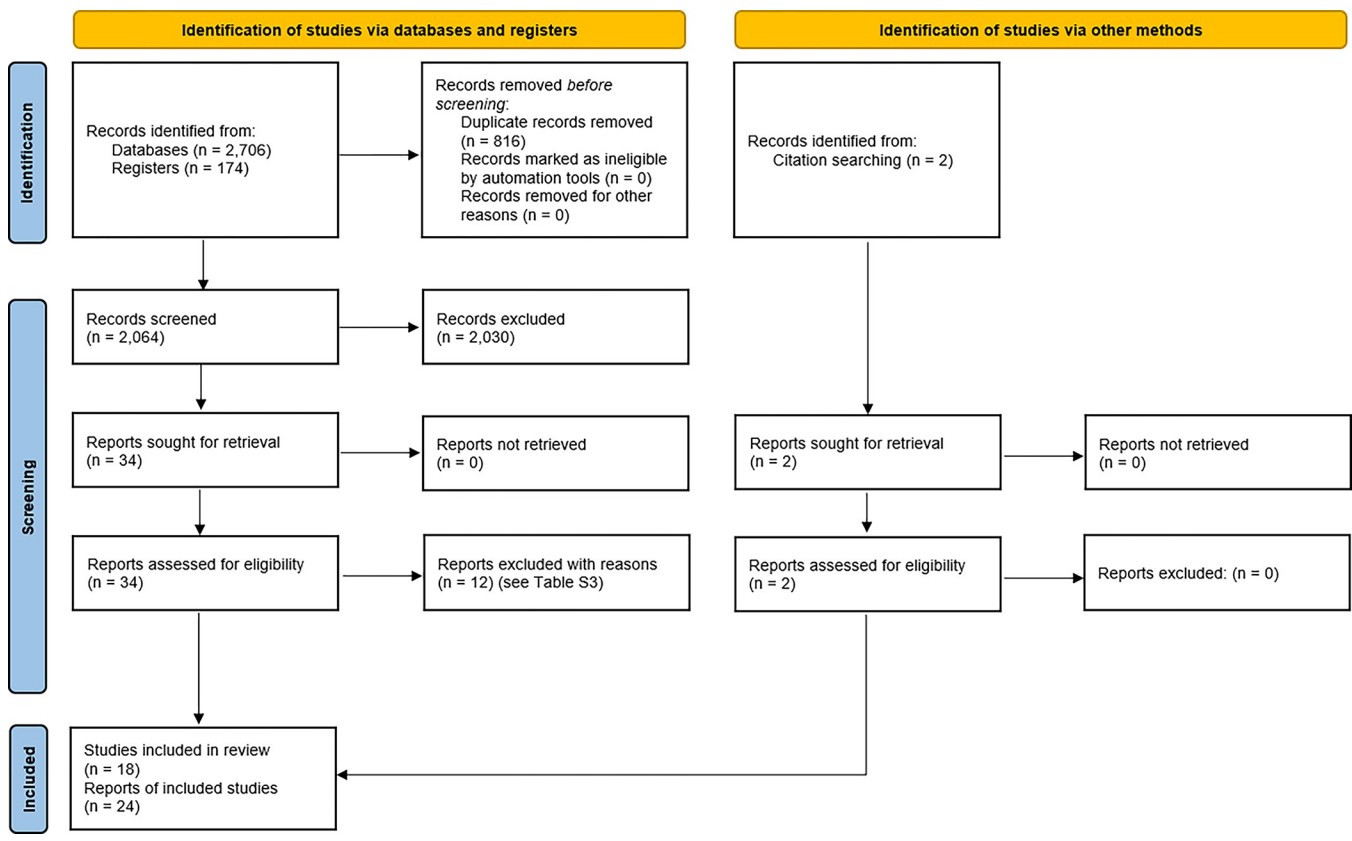

**Fig 1. PRISMA flow diagram of the study selection process.**

## 3.2. Risk of bias

Among randomized studies (S4 Table), seven [50,54,56,59,62,63,65] were judged as low RoB, while the remaining six were rated as high RoB [43,45,46] or some concerns for bias due to randomization [42,48,60].

Among five non-randomized studies (S5 Table), only one trial by Kazmin et al. [61] was judged as low RoB in all dimensions. In the other four studies [44,47,52,53], there was a moderate-to-high RoB due to confounding and participant selection issues.

## 3.3. Immunogenicity towards vaccine-like strains

Findings on the HAI response against homologous vaccine-like strains approximately 3–4 weeks after vaccination were reported in 17 publications [42,44–48,50,52,54,56,59–65]. As shown by the extracted raw data (S6 Table), strain-specific SCRs and SPRs following one dose of aTIV/aQIV exceeded 40% and 70%, respectively, in the large majority of trials. For relative immunogenicity parameters, aTIV/aQIV was generally more immunogenic than non-adjuvanted counterparts. However, statistically significant differences were observed only in some, generally larger studies. In these latter trials, the advantage of aTIV/aQIV over TIV/QIV was more pronounced for type A strains. For GMTRs (aTIV/aQIV vs. TIV/QIV), use of aTIV/aQIV was associated with higher GMTs (GMTR >1.00) in 39 out of 43 (91%) estimates extracted, where the relative advantage of aTIV/aQIV varied from 6% to 250%. However, only 20 estimates were statistically significant (p<0.05) (S6 Table).

**Table 1. Characteristics of the studies included.**

| Study [Ref] | Study design | Outcomes of interest reported | Country (season) | Study population | Adjuvanted vaccine (N) | Comparator (N) |
|---|---|---|---|---|---|---|
| Frey 2003 [42] | Randomized, observer-blind, controlled, multicenter | HAI (vaccine-like strains), safety | United States (1995/1996) | 18–64 years (healthy) | aTIV (150) | TIV (151) |
| Magnani 2005 [43] | Randomized, open-label, controlled, single-center | ELISA | Italy (1999/2000) | Heart transplant recipients (mean age 55 years) | aTIV (21) | TIV (21), no vaccination (16) |
| Lorio 2003 [44] | Systematic allocation, controlled, single-center | HAI (vaccine-like strains) | Italy (2000/2001) | Non-elderly adults (HIV-1-seropositive) | aTIV (44) | TIV (40) |
| Pollok 2004 [45] | Randomized, observer-blind, controlled, multicenter | HAI (vaccine-like strains), safety | Germany (2000/2001) | 18–64 years (renal transplant recipients) | aTIV (60) | TIV (56) |
| Gabutti 2005 [46] | Randomized, open-label, controlled, multicenter | HAI (vaccine-like strains), safety | Italy (2002/2003) | 18–65 years (HIV-1-seropositive) | aTIV (18) | TIV (19) |
| Camilloni 2009 [47], Iorio 2011 [64] | Non-randomized (single-arm), single-center | HAI (vaccine-like and heterologous strains) | Italy (2003/2004) | 32–46 years (healthy) | aTIV (26) | None |
| Durando 2008 [48], Fenoglio 2011 [49] | Randomized, open-label, controlled, multicenter | HAI (vaccine-like strains), cell-mediated immunity, safety | Italy (2005/2006) | 18–65 years (37.1% positive for HIV-1) | aTIV (127) | TIV (129) |
| Baldo 2007 [50,51] | Randomized, double-blind, controlled, single-center | HAI (vaccine-like and heterologous strains), safety | Italy (2005/2006) | 18–60 years (with comorbidities) | aTIV (128) | TIV (128) |
| Baldo 2012 [65] | Phase III, randomized, controlled, observer-blind, single-center | HAI (vaccine-like and heterologous strains), safety | Italy (2006/2007) | 18–60 years (with comorbidities) | aTIV (180) | TIV (179) |
| Lorio 2012 [52] | Non-randomized (single-arm), single-center | HAI (vaccine-like and heterologous strains), cell-mediated immunity | Italy (2007/2008) | 25–56 years (healthy) | aTIV (17) | None |
| Fabbiani 2013 [53] | Non-randomized (single-arm), single-center | ELISA, cell-mediated immunity | Italy (2010/2011) | Adults [median (IQR) age 45 (34–53) years] (73.0% positive for HIV) | aTIV (111) | None |
| Kumar 2016 [54] and 2017 [55] | Randomized, double-blind, controlled, single-center | HAI (vaccine-like and heterologous strains), safety | Canada (2012/2013) | Adult kidney transplant recipients (83% were 18–64 years) | aTIV (34) | TIV (34) |
| Spensieri 2016 [56], de Wolf 2017 [57], Weiner 2019 [58] | Exploratory, randomized, partially blinded (vaccinee and laboratory staff), controlled, single-center | HAI (vaccine-like strains), cell-mediated immunity | United Kingdom (2012/2013) | 18–45 years (healthy) | aTIV (21) | TIV (21), placebo (8) |
| Noh 2016 [59] | Randomized, open-label, controlled, multicenter | HAI (vaccine-like strains) | Korea (2013/2014) | 19–64 years (patients with chronic kidney disease and hemodialysis) | aTIV (67) | TIV (58) |
| Natori 2017 [60] | Randomized, blinded, controlled, single-center | HAI (vaccine-like strains), safety | Canada (2015/2016) | Adult allogeneic hematopoietic stem cell transplant recipients (median age 53.5 years) | aTIV (35) | TIV (38) |
| Kazmin 2023 [61] | Phase II, single-arm, open-label, single center | HAI (vaccine-like strains), cell-mediated immunity, safety | United Kingdom (2015/2016) | 25–40 years (healthy) | aTIV (31) | None |
| Mombelli 2024 [62] | Randomized, double-blind, controlled, superiority multicenter | HAI (vaccine-like strains), efficacy, safety | Switzerland, Spain (2018/2019, 2019/2020) | Adult solid organ transplant recipients [median (IQR) age 57 (45–64) years] | aTIV (209) | QIV (204), hdTIV (203) |
| Poder 2023 [63] | Randomized, observer-blind, controlled, multicenter | HAI (vaccine-like strains), safety | Estonia, Germany, United States (2021/2022) | 50–64 years (89.6% healthy) | aQIV (1,027) | QIV (1,017) |

aQIV, Quadrivalent MF59-adjuvanted seasonal influenza vaccine; aTIV, Trivalent MF59-adjuvanted seasonal influenza vaccine; ELISA, Enzyme-linked immunosorbent assay; HAI, Hemagglutination inhibition assay; hdTIV, High-dose trivalent seasonal influenza vaccine; HIV, Human immunodeficiency virus; IQR, Interquartile range; QIV, Quadrivalent non-adjuvanted seasonal influenza vaccine; TIV, Trivalent non-adjuvanted seasonal influenza vaccine.

Depending on the vaccine strain and immunological parameter, pooled analysis could be performed on 14–17 vaccine cohorts. RE and FE pooled estimates were generally similar in terms of the effect size, but RE estimates were less precise. As expected, prevalence estimates of the absolute SCRs and SPRs showed high heterogeneity ($\geq$82%) (Table 2, S1–S6 Figs). Pooled RE estimates for SCRs were 56.5%, 62.8% and 53.5% towards A(H1N1), A(H3N2) and B vaccine-like strains, respectively, and all lower bounds of the related 95% CIs exceeded 40%. The corresponding pooled RE estimates for SPRs (HAI titer $\geq$1:40) were 89.7%, 90.7% and 80.8%, and lower bounds of their 95% CIs were $\geq$70%. As one RCT [42] defined SPR using a more conservative HAI titer threshold of $\geq$1:160, a sensitivity analysis was conducted by also including this study. No significant changes occurred and the RE estimates were 90.7%, 90.9%, and 82.7% towards A(H1N1), A(H3N2) and B vaccine-like strains, respectively (S7 Table).

Meta-analytical estimates for the relative immunogenicity parameters were generally associated with less heterogeneity (Table 2, S7–S12 Figs). For the A(H1N1) strain, 8.8% (95% CI: 3.7%, 14.0%; $I^2$ = 49.1%) more aTIV/aQIV than TIV/QIV recipients seroconverted, while the difference in SPRs was 3.8% (95% CI: 1.0%, 6.6%; $I^2$ = 44.3%). The advantage of aTIV/aQIV was more pronounced for the A(H3N2) subtype, where RE $\Delta$SCR and $\Delta$SPR were 13.1% (95% CI: 6.7%, 19.6%; $I^2$ = 69.9%) and 8.6% (95% CI: 4.0%, 13.2%; $I^2$ = 74.3%), respectively. For the B vaccine-like strains, aTIV/aQIV determined higher SCRs (11.7%; 95% CI: 7.2%, 16.2%; $I^2$ = 36.3%) and SPRs (5.1%; 95% CI: 1.6%, 8.6%; $I^2$ = 44.3%) than TIV/QIV (Table 2).

In a pre-specified subgroup analysis, immunosuppression status was a significant determinant of most pooled estimates. As shown in Table 3, compared with trials on general adult populations, individuals with immunosuppressive conditions showed significantly lower SCRs and SPRs. For instance, the RE estimates for SCRs towards A(H1N1) among individuals with and without immunosuppression were 47.1% and 65.0%, respectively. However, when considering relative immunogenicity parameters, the advantage of aTIV/aQIV was more pronounced in immuno-compromised populations. For example, $\Delta$SCR aTIV/aQIV vs. TIV/QIV towards A(H1N1) were 13.1% and 5.4% in adults with and without immunosuppression, respectively (Table 3).

Visual inspection of funnel plots (S13–S18 Figs) suggested some evidence of publication bias, especially for $\Delta$SCR and $\Delta$SPR towards A(H3N2). In a sensitivity analysis with the trim-

**Table 2. Meta-analysis of absolute and relative seroconversion and seroprotection rates towards vaccine-like strains 3–4 weeks after one dose of the MF59-adjuvanted or non-adjuvanted seasonal influenza vaccines in non-elderly adults, by vaccine strain.**

| Parameter | Vaccine-like strain | k | $I^2$, % | FE model, % (95% CI) | RE model, % (95% CI) |
|---|---|---|---|---|---|
| SCR | A(H1N1) | 17 | 94.7 | 67.1 (65.0, 69.1) | 56.5 (48.7, 64.1) |
| | A(H3N2) | 16 | 82.2 | 63.6 (61.5, 65.8) | 62.8 (55.6, 69.8) |
| | B | 16 | 92.3 | 48.6 (46.4, 50.8) | 53.5 (44.1, 62.7) |
| SPR | A(H1N1) | 15 | 94.1 | 97.2 (96.3, 98.0) | 89.7 (82.9, 95.1) |
| | A(H3N2) | 15 | 92.8 | 94.3 (93.1, 95.3) | 90.7 (84.3, 95.7) |
| | B | 16 | 95.2 | 86.9 (85.3, 88.4) | 80.8 (72.0, 88.4) |
| $\Delta$SCR | A(H1N1) | 14 | 49.1 | 7.1 (4.4, 9.9) | 8.8 (3.7, 14.0) |
| | A(H3N2) | 14 | 69.9 | 8.1 (5.1, 11.0) | 13.1 (6.7, 19.6) |
| | B | 14 | 36.3 | 8.5 (5.6, 11.5) | 11.7 (7.2, 16.2) |
| $\Delta$SPR | A(H1N1) | 14 | 52.5 | 3.3 (1.8, 4.8) | 3.8 (1.0, 6.6) |
| | A(H3N2) | 14 | 74.3 | 5.3 (3.5, 7.1) | 8.6 (4.0, 13.2) |
| | B | 14 | 44.3 | 4.6 (2.6, 6.7) | 5.1 (1.6, 8.6) |

FE, Fixed effects; k, Number of studies; RE, Random effects; SCR, Seroconversion rate; SPR, Seroprotection rate; $\Delta$SCR: Difference in seroconversion rates between individuals immunized with adjuvanted vs non-adjuvanted influenza vaccines; $\Delta$SPR: Difference in seroprotection rates between individuals immunized with adjuvanted vs non-adjuvanted influenza vaccines.

**Table 3. Meta-analysis of absolute and relative seroconversion and seroprotection rates towards vaccine-like strains 3–4 weeks after one dose of the MF59-adjuvanted or non-adjuvanted seasonal influenza vaccines in non-elderly adults: A subgroup analysis by the presence of immunosuppressive conditions.**

| Parameter | Vaccine-like strain | Immunosuppression | k | $I^2$, % | FE model, % (95% CI) | RE model, % (95% CI) |
|---|---|---|---|---|---|---|
| SCR | A(H1N1) | No | 9 | 89.8 | 74.6 (72.4, 76.8) | 65.0 (57.6, 72.0) |
| | | Yes | 8 | 85.6 | 40.2 (35.7, 44.7) | 47.1 (35.6, 58.9) |
| | A(H3N2) | No | 8 | 83.5 | 66.7 (64.2, 69.0) | 70.3 (61.1, 78.7) |
| | | Yes | 8 | 57.4 | 53.5 (48.9, 58.0) | 54.4 (46.4, 62.4) |
| | B | No | 8 | 93.2 | 51.9 (49.4, 54.5) | 60.2 (49.6, 70.3) |
| | | Yes | 8 | 89.2 | 38.1 (33.7, 42.6) | 46.5 (32.4, 61.0) |
| SPR | A(H1N1) | No | 7 | 92.1 | 99.5 (98.9, 99.9) | 96.3 (90.0, 99.8) |
| | | Yes | 8 | 66.4 | 83.4 (79.8, 86.7) | 82.1 (74.3, 88.9) |
| | A(H3N2) | No | 7 | 90.7 | 96.9 (95.8, 97.9) | 94.6 (87.5, 99.1) |
| | | Yes | 8 | 86.8 | 82.8 (79.2, 86.2) | 86.4 (76.1, 94.4) |
| | B | No | 8 | 95.1 | 91.1 (89.4, 92.6) | 83.5 (71.3, 92.9) |
| | | Yes | 8 | 91.3 | 70.9 (66.7, 75.0) | 78.1 (64.4, 89.4) |
| ΔSCR | A(H1N1) | No | 6 | 34.3 | 5.4 (2.2, 8.5) | 5.4 (1.3, 9.5) |
| | | Yes | 8 | 41.9 | 12.6 (7.0, 18.3) | 13.1 (4.6, 21.5) |
| | A(H3N2) | No | 6 | 65.7 | 4.7 (1.3, 8.1) | 7.5 (-0.2, 15.1) |
| | | Yes | 8 | 31.2 | 18.8 (12.8, 24.8) | 19.7 (12.3, 27.1) |
| | B | No | 6 | 34.2 | 6.5 (3.0, 9.9) | 8.4 (2.8, 14.0) |
| | | Yes | 8 | 0.0 | 15.0 (9.7, 20.4) | 15.4 (10.1, 20.7) |
| ΔSPR | A(H1N1) | No | 6 | 63.6 | 2.5 (1.3, 3.7) | 3.4 (0.1, 6.8) |
| | | Yes | 8 | 33.5 | 6.1 (1.1, 11.0) | 5.0 (0.2, 9.7) |
| | A(H3N2) | No | 6 | 80.4 | 3.4 (1.7, 5.0) | 6.6 (-0.1, 13.4) |
| | | Yes | 8 | 26.8 | 11.6 (6.4, 16.9) | 10.4 (5.8, 14.9) |
| | B | No | 6 | 64.3 | 3.9 (1.8, 5.9) | 5.8 (0.3, 11.3) |
| | | Yes | 8 | 2.0 | 7.2 (1.7, 12.7) | 5.8 (0.7, 10.9) |

FE, Fixed effects; k, Number of studies; RE, Random effects; SCR, Seroconversion rate; SPR, Seroprotection rate; ΔSCR: Difference in seroconversion rates between individuals immunized with adjuvanted vs. non-adjuvanted influenza vaccines; ΔSPR: Difference in seroprotection rates between individuals immunized with adjuvanted vs. non-adjuvanted influenza vaccines.

and-fill method, the number of hypothetically missing studies ranged from zero to six. With these imputed studies, RE estimates for the difference in SCRs were 8.8% (95% CI: 3.7%, 14.0%), 3.3% (95% CI: -4.9%, 11.5%), and 7.4% (95% CI: 2.3%, 12.5%) towards A(H1N1), A(H3N2), and B, respectively. The corresponding ΔSPR parameters were 1.4% (95% CI: -2.8%, 5.6%), 1.7% (95% CI: -4.8%, 8.2%), and 2.1% (95% CI: -2.5%, 6.8%), respectively.

When three studies [54,60,62] with partially overlapping populations were excluded in another sensitivity analysis (S8 Table), no major changes occurred.

In an exploratory meta-regression analysis to investigate study-level determinants of ΔSCRs and ΔSPRs for aTIV/aQIV vs. TIV/QIV (S9 Table), only one statistically significant association emerged. Particularly, studies on immunocompromised populations showed, on average, greater (p = 0.030) ΔSCRs towards A(H3N2). No significant associations for other variables, including below the median sample size, industry sponsorship, RoB, and enrollment of overlapping population groups, were found.

As several studies did not report dispersion parameters for the GMT point estimates, their imputation was judged unfeasible (as it could favor positive results). Meta-analysis of GMTs for aTIV/aQIV vs. TIV/QIV was therefore abandoned.

### 3.4. Immunogenicity towards heterologous strains

HAI immune response towards heterologous strains was assessed in six publications [47,51,52,55,64,65] (S10 Table). Three of these studies [47,52,64] were small single-arm trials. Camilloni et al. [47] analyzed post-vaccination SPRs in adults vaccinated with the 2002/2003 aTIV formulation that contained a B strain belonging to the Victoria lineage (B/Hong Kong/330/2001). Post-vaccination sera were tested against three different heterologous B strains, one of which belonged to the same Victoria lineage (B/Malaysia/2506/2004) and the other two belonged to the Yamagata lineage (B/Sichuan/379/1999 and B/Shanghai/361/2002). One month after vaccination with aTIV, a high level of response was observed for the same-lineage strain and one cross-lineage strain B/Sichuan/379/1999 with 92.3% and 88.5% of individuals achieving HAI titers ≥1:40, respectively. Conversely, the derived SPR towards another cross-lineage strain B/Shanghai/361/2002 was relatively low (38.5%). Iorio et al [64] analyzed humoral immunogenicity in adults vaccinated with aTIV formulations containing seasonal (pre-2009) A(H1N1) strains (formulations 2003/2004 [64] and 2007/2008 [52]) towards an A(H1N1)pdm09-like strain. SCRs and SPRs were 12.5–19.2% and 12.5–30.7% (S10 Table).

Three RCTs [51,55,65] compared heterologous immune response of aTIV vs. TIV. The former was usually associated with higher immune response (S10 Table). In the pooled analysis of these studies (S11 Table), aTIV was associated with significantly higher cross-clade SCRs (FE: 10.7% [95% CI: 3.2%, 18.2%); RE: 10.6% [95% CI: 3.2%, 18.0%]; $I^2 = 0\%$] and SPRs (FE: 10.5% [95% CI: 4.9%, 16.1%]; RE: 10.2% [95% CI: 0.5%, 19.9%]; $I^2 = 55.6\%$] towards A(H3N2) strains. For the A(H1N1) heterologous strains, aTIV was associated with higher SPRs (FE: 10.0% [95% CI: 0.7%, 19.3%]; RE: 9.0% [95% CI: 0.1%, 17.9%]; $I^2 = 0\%$], but not SCRs (FE: 8.3% [95% CI: -1.2%, 17.8%]; RE: 0.2% [95% CI: -26.2%, 26.2%]; $I^2 = 83.4\%$]. The difference between aTIV and TIV was not significant for the cross-lineage B response.

In all available studies, aTIV determined higher GMTs (by 3–90%) for all heterologous strains; 50% (4/8) of these estimates were statistically significant. Meta-analysis of the differences in GMTs was not performed, as no studies reported dispersion parameters.

### 3.5. Persistence of antibodies

In seven studies [42,46,48,56,59,62,63] reporting immunogenicity findings at longer post-vaccination periods (mostly 6 months), 43–100% of aTIV/aQIV users were still seroprotected. Although SPRs were generally higher in aTIV/aQIV than TIV/QIV users, the reported differences were usually not statistically significant (S12 Table).

In the pooled analysis of absolute SPRs at 6 months (S13 Table), the RE estimates ($I^2 > 96\%$) were 82.0% (95% CI: 63.4%, 95.3%), 80.5% (95% CI: 63.0%, 93.6%), and 65.4% (95% CI: 47.4%, 81.5%) towards A(H1N1), A(H3N2), and B strains, respectively. When compared with TIV/QIV, aTIV/aQIV was associated with marginally higher SPRs at six months towards A(H1N1) (FE: 4.1% [95% CI: 2.1%, 6.2%]; RE: 6.9% [95% CI: -0.2%, 14.0%]; $I^2 = 65.7\%$) and A(H3N2) (FE: 3.5% [95% CI: 1.0%, 6.1%]; RE: 4.2% [95% CI: -0.3%, 8.6%]; $I^2 = 0\%$) but not influenza B (FE: 1.2% [95% CI: -1.8%, 4.2%]; RE: 3.3% [95% CI: -4.2%, 10.8%]; $I^2 = 43.0\%$).

### 3.6. Cell-mediated immunogenicity

Findings on cell-mediated immunity were reported in eight publications [48,49,52,53,56–58,61]. As expected, these trials used different assays and methods and therefore were summarized narratively. In the RCT by Durando et al. [48], double-positive CD3+CD4+ T-cell proliferation was assessed 1 and 3 months after a dose of aTIV or TIV was administered to HIV-1-seronegative and HIV-1-seropositive adults. In HIV-1-seronegative adults, both vaccines induced a measurable increase in memory T lymphocytes at both time points. However, the

difference between the two vaccine arms was not statistically significant. By contrast, 1 month after vaccination, there was a clear advantage (p = 0.0002) of aTIV in HIV-1-seropositive adults. A subsequent subset study [49], compared production of interleukins (IL) 23 (IL-23) and IL-6 in response to stimulus with hemagglutinin. Both IL-6 and IL-23 syntheses increased (p≤0.001) upon vaccination with aTIV but not TIV. This increase was seen in both HIV-1 negative and positive cohorts. Conversely, Fabbiani et al. [53] reported that following one dose of aTIV, the production of cytokines increased significantly only in HIV-negative individuals. Among HIV-positive individuals, higher HIV viral load was significantly associated with reduced post-vaccination IL-10 levels.

A small single-arm study by Iorio et al. [52] showed that aTIV induced antigen-specific activation of T-cell responses, which were measured through quantification of double-positive CD69+CD3+ or CD69+CD8+ lymphocytes. A significant increase was shown for all vaccine-like strains and a heterologous A(H1N1)pdm09-like strain (aTIV contained a pre-2009 seasonal H1N1 strain). A similar increase was also seen in the interferon-γ enzyme-linked immunospot assay. The authors also reported no correlation between cellular and humoral immunogenicity of aTIV [52].

In the BIOVACSAFE study [56–58] the frequency of H1N1-specific IL-21 producing T follicular helper cells persisted at higher levels in aTIV recipients compared with adults vaccinated with TIV [56]. However, both TIV and aTIV did not result in significant changes in frequencies and phenotypes of resting and activated regulatory T-cells [57]. aTIV induces significantly high early activation of interferon- and innate-cell-related genes, which are indispensable for control of viral infections and immune regulation [58]. Finally, vaccination with aTIV in adults was shown to induce a persistent transcriptional signature of innate immunity [61].

### 3.7. Immune response measured in ELISA

Two studies [43,53] reported results of aTIV-induced immunogenicity measured by commercially available ELISA kits. These data should be interpreted cautiously since this assay is not recommended by regulatory bodies [27] and may be associated with inaccurate results [66]. Magnani et al. [43] reported that, among heart transplant recipients, both aTIV and TIV determined a significant rise in both IgM and IgG towards both influenza A and B. However, there was no difference between the two vaccines. A single-arm study by Fabbiani et al. [53] showed a significant increase in IgM and IgG in both HIV-positive and HIV-negative adults. Notably, the post-vaccination fold-rise in IgM was higher in HIV-positive than in HIV-negative individuals (4.35 vs. 1.14).

### 3.8. Efficacy

Efficacy against RT-PCR-confirmed influenza was a secondary outcome in an RCT that enrolled solid-organ transplant recipients [62]. Participants were randomized on a 1:1:1 ratio to receive QIV, aTIV or high dose non-adjuvanted TIV (hdTIV). The incidence of influenza in the study arms was similar: 5.6% (11/198), 5.4% (11/205) and 6.7% (13/195) of participants in QIV, aTIV and hdTIV groups, respectively, were diagnosed with laboratory-confirmed influenza. When restricted to the active surveillance RT-PCR (i.e., subjects self-collected five nasopharyngeal swabs at weeks 2, 4, 6, 8 and 10 after the start of the influenza season), the incidence of influenza was 5.1% (10/198), 3.4% (7/205) and 4.6% (9/195) in QIV, aTIV and hdTIV arms, respectively. However, it should be stressed that this RCT was not powered to find differences in the efficacy estimates, as its primary outcome was humoral immune response.

**Table 4.** Meta-analysis of absolute and relative solicited reactogenicity during the first week after one dose of the MF59-adjuvanted or non-adjuvanted seasonal influenza vaccines in non-elderly adults, by adverse reaction.

| Adverse reaction | | k | $I^2$, % | FE model | RE model |
|---|---|---|---|---|---|
| **Absolute frequency of adverse reactions in adults vaccinated with aTIV/aQIV, % (95% CI)** | | | | | |
| Local | Any | 4 | 18.5 | 49.2 (46.5, 51.9) | 49.2 (46.5, 51.9) |
| | Pain | 9 | 95.6 | 49.9 (47.7, 52.2) | 51.0 (35.6, 66.4) |
| | Erythema | 7 | 78.8 | 8.7 (7.4, 10.0) | 9.8 (6.3, 14.0) |
| | Induration | 6 | 85.9 | 10.3 (8.8, 11.8) | 13.1 (8.5, 18.6) |
| | Ecchymosis | 4 | 89.2 | 1.2 (0.7, 1.9) | 2.7 (0.5, 6.3) |
| Systemic | Any | 4 | 91.6 | 43.2 (40.5, 45.9) | 31.8 (14.4, 52.3) |
| | Fever | 10 | 89.6 | 2.7 (1.9, 3.6) | 3.5 (0.6, 8.0) |
| | Chills | 4 | 0.0 | 6.6 (5.3, 7.9) | 6.6 (5.3, 7.9) |
| | Myalgia | 7 | 82.9 | 14.5 (12.9, 16.1) | 15.3 (10.7, 20.5) |
| | Arthralgia | 6 | 91.6 | 11.4 (9.9, 12.9) | 9.0 (3.8, 15.9) |
| | Headache | 6 | 78.7 | 20.6 (18.8, 22.5) | 19.4 (14.3, 25.1) |
| | Malaise | 4 | 87.6 | 19.7 (16.6, 23.1) | 19.9 (11.4, 30.2) |
| | Nausea | 4 | 63.4 | 6.1 (4.9, 7.4) | 5.1 (3.0, 7.7) |
| **Relative risk (RR) of adverse reactions in adults vaccinated with aTIV/aQIV vs. TIV/QIV, RR (95% CI)** | | | | | |
| Local | Any | 4 | 0.0 | 1.69 (1.53, 1.86) | 1.68 (1.52, 1.86) |
| | Pain | 8 | 69.4 | 1.80 (1.66, 1.95) | 1.93 (1.60, 2.32) |
| | Erythema | 7 | 70.1 | 1.59 (1.26, 2.01) | 1.49 (0.91, 2.42) |
| | Induration | 6 | 16.1 | 1.82 (1.44, 2.30) | 1.75 (1.30, 2.35) |
| | Ecchymosis | 4 | 0.0 | 1.08 (0.63, 1.85) | 1.06 (0.61, 1.83) |
| Systemic | Any | 4 | 68.7 | 1.21 (1.10, 1.33) | 1.35 (1.01, 1.81) |
| | Fever | 8 | 27.9 | 2.24 (1.54, 3.28) | 2.07 (1.14, 3.78) |
| | Chills | 4 | 17.2 | 1.21 (0.91, 1.61) | 1.18 (0.88, 1.57) |
| | Myalgia | 7 | 53.1 | 1.89 (1.57, 2.29) | 1.94 (1.37, 2.75) |
| | Arthralgia | 6 | 0.0 | 1.49 (1.22, 1.81) | 1.47 (1.21, 1.79) |
| | Headache | 6 | 0.0 | 1.15 (1.01, 1.31) | 1.15 (1.00, 1.31) |
| | Malaise | 4 | 0.0 | 1.59 (1.23, 2.07) | 1.60 (1.23, 2.08) |
| | Nausea | 4 | 0.0 | 1.59 (1.16, 2.18) | 1.59 (1.16, 2.19) |

aQIV, Quadrivalent MF59-adjuvanted seasonal influenza vaccine; aTIV, Trivalent MF59-adjuvanted seasonal influenza vaccine; FE, Fixed effects; k, Number of studies; QIV, Quadrivalent non-adjuvanted seasonal influenza vaccine; RE, Random effects; RR, Relative risk; TIV, Trivalent non-adjuvanted seasonal influenza vaccine.

## 3.9. Safety, reactogenicity and tolerability

At least one safety aspect of interest was reported in 11 trials [42,45,46,48,50,54,60–63,65]. Compared with non-adjuvanted vaccines, aTIV/aQIV was associated with an increased risk of solicited reactions, especially those at the injection site (S14 Table). However, the overwhelming majority of the reactogenic events were mild-to-moderate and self-limiting.

In the pooled RE analysis, any local and systematic reactions were reported by 49.2% and 31.8% of adults vaccinated with aTIV/aQIV. The most common injection site reactions were pain (51.0%) and induration (13.1%), while malaise (19.9%), headache (19.4%), and myalgia (15.3%) were the most common systemic reactions. Other solicited reactions were less frequent (<10%) (Table 4, S19–S31 Figs). Pooled analysis for rash was not performed, as no cases (0%) were reported in two studies [42,50].

In the pooled analysis of relative (vs. TIV/QIV) reactogenicity, aTIV/aQIV users showed increased RRs of any local (FE RR: 1.69; $I^2$ = 0%) and systemic (RE RR: 1.35; $I^2$ = 68.7%) reactions, pain (RE RR: 1.93; $I^2$ = 69.4%), induration (FE RR: 1.82; $I^2$ = 16.1%), fever (FE RR: 2.24;

$I^2$ = 27.9%), myalgia (RE RR: 1.94; $I^2$ = 53.1%), arthralgia (FE RR: 1.49; $I^2$ = 0%), headache (FE RR: 1.15; $I^2$ = 0%), malaise (FE RR: 1.59; $I^2$ = 0%), and nausea (FE RR: 1.59; $I^2$ = 0%) (**Table 4**, **S32–S44 Figs**).

As for unsolicited events, SAEs were infrequent (FE: 1.6% [95% CI: 0.9%, 2.4%]; RE: 0.8% [95% CI: 0.0%, 2.3%]; $I^2$ = 62.0%) (**S45 Fig**) and most of them were unrelated to study vaccines. There was no difference in SAE reporting between aTIV/aQIV and TIV/QIV arms (FE RR: 1.02 [95% CI: 0.64, 1.63]; RE: 1.01 [95% CI: 0.63, 1.62]; $I^2$ = 0%) (**S46 Fig**).

## 4. Discussion

This review systematically appraised available experimental evidence on the performance of aTIV/aQIV in non-elderly adults. We showed that aTIV/aQIV is immunogenic and satisfies the currently available regulatory criteria for the absolute immune response [27]. Some evidence suggests that aTIV/aQIV is also immunogenic against antigenically dissimilar strains, which may be indicative of a certain level of cross-protection. Compared with non-adjuvanted standard-dose vaccines, aTIV/aQIV may be more immunogenic, although the magnitude of effect size depends on immunological parameter, vaccine strain and presence immunosuppressive conditions. Despite some increase in transient reactogenic events, aTIV/aQIV has an acceptable safety profile and the incidence of SAEs is low and comparable to that observed with non-adjuvanted vaccines. However, data on efficacy of aTIV/aQIV are scant and inconclusive. In view of its recent expansion in terms of age indication, which changed from ≥65 to ≥50 years [15], aTIV/aQIV is a valuable option for the prevention of seasonal influenza in adults aged 50–64 years, especially those who are immunocompromised. However, the current evidence does not allow inference of any preference of aTIV/aQIV over non-adjuvanted vaccines.

From its first Italian registration in 1997, experimental and observation research on aTIV/aQIV has been mostly focused on older adults [1–6]. It is therefore worth comparing our findings with those reported by the latest immunogenicity meta-analysis by Nicolay et al. [4], which is based on 39 studies involving 27,116 individuals aged ≥65 years. In their study, at 1 month post-vaccination, the absolute difference in SCRs for aTIV vs. TIV towards vaccine-like strains was 9.5% (95% CI: 5.2%, 13.9%), 10.5% (95% CI: 6.6%, 14.5%), and 12.7% (95% CI: 8.6%, 16.8%) against A(H1N1), A(H3N2) and B strains, respectively. In our study, the corresponding differences established in RE models were comparable: 8.8% (95% CI: 3.7%, 14.0%), 13.1% (95% CI: 6.7%, 19.6%) and 11.7% (95% CI: 7.2%, 16.2%), respectively. Regarding differences in SPRs, similarly to our findings, Nicolay et al. [4] reported smaller but still significant effect sizes (A[H1N1]: 2.4% [95% CI: 0.8%, 4.0%]; [(H3N2]: 2.7% [95% CI: 0.9%, 4.5%]; B: 4.5% [95% CI: 1.8%, 7.1%]). Collectively, these data indicate that aTIV/aQIV is more immunogenic than non-adjuvanted standard-dose vaccines in both elderly and non-elderly adults.

We also demonstrated that, compared with general adult cohorts, the advantage of aTIV over TIV was greater in immunocompromised individuals. This superiority was particularly pronounced for the A(H3N2) subtype, where the absolute difference in SCRs reached approximately 20%. An early meta-analysis (13 studies) by Banzhoff et al. [3] compared the immunogenicity of aTIV and TIV in older adults with or without underlying chronic conditions. In healthy adults, GMTRs for aTIV vs. TIV were 1.10, 1.18 and 1.17 towards A(H1N1), A(H3N2) and B strains, respectively. The corresponding GMTRs in adults with co-morbidities were higher: 1.17, 1.43 and 1.37, respectively. When the presence of co-morbidities was added as a covariate, their model showed a significant interaction effect (p = 0.004) for A(H3N2), while this latter was not significant for A(H1N1) (p = 0.41) and B (p = 0.065) strains. The most probable explanation for this finding is poor immunogenicity of standard influenza vaccines in

individuals with immunodeficiencies [26,67] and therefore an extra effort is needed to achieve protective titers in this target group [68]. Additionally, limited evidence suggests [48,49] that compared with TIV, aTIV enhances some components of cell-mediated immunity only in adults with immunodeficiency. Altogether, these data suggest that individuals with underlying immunosuppressive conditions may benefit more from aTIV/aQIV and this population group may be considered a primary target for the recent aQIV availability for adults aged 50–64 years.

Our third major finding is that aTIV is statistically superior to TIV in terms of cross-clade immunogenicity towards the A(H3N2) subtype: the absolute difference in both SCRs and SPRs was approximately 10%. This result is fully in line with several previous studies on heterologous immune response induced by aTIV in the elderly [4,69–71]. In older adults, aTIV has been also shown to be more effective than QIV in preventing hospitalizations due to influenza A(H3N2) [72]. Mechanistically, non-adjuvanted vaccines mostly recognize epitopes of the more conserved stem region of hemagglutinin. Conversely, MF59 induces epitope spreading from hemagglutinin stem to hemagglutinin globular head, which is much more variable and subjected to immune pressure generating antibody escape variants [73]. Indeed, the A(H3N2) subtype has the highest mutation rate [74] and vaccine effectiveness against this virus is comparatively low in all age groups [75]. However, in view of the absence of efficacy/effectiveness data in non-elderly adults and considering that HAI is an imperfect correlate of protection [25,26], it remains unclear whether the observed immunogenicity advantage of aTIV can translate into a better protection against influenza A(H3N2) in non-elderly adults.

The available data on antibody persistence are limited and less conclusive. While at 6 months, the period which roughly corresponds to the entire influenza season, most individuals vaccinated with aTIV were still seroprotected, the advantage of aTIV/aQIV over TIV/QIV diminished and most pooled estimates were not statistically significant. Similar immunogenicity results have been established in a large pivotal RCT of aTIV vs. TIV in the elderly [76]. In that trial, GMTRs for aTIV vs. TIV were maximum at 1 month post-vaccination and then approached 1.0 (i.e. no difference between the two vaccines) at later time periods.

In adults, the use of aTIV/aQIV may be judged safe, as most AEs are reactogenic in nature, and transient and mild-to-moderate in intensity, while SAEs are uncommon. However, aTIV/aQIV was generally more reactogenic compared to TIV/QIV, especially when considering local injection site reactions. The observed effect sizes (RRs of 1.69 and 1.35 for any local and any systemic events, respectively) were similar to those previously reported. For example, the latest meta-analysis by O Murchu [36], which predominantly included studies on older adults, has reported pooled RRs of 1.90 (95% CI: 1.50, 2.39) and 1.18 (95% CI: 1.02, 1.38) for any local and any systemic reactions, respectively. Being immunostimulants, all adjuvants usually increase reactogenicity compared with inactivated formulations without adjuvant [77,78]. On the contrary, the incidence of unsolicited SAEs, most of which were unrelated to study vaccines, was similar between aTIV/aQIV and TIV/QIV. This is, again, in line with the previous pooled analyses on older adults [79,80].

We identified some important study limitations. At review level, we were unable to identify unpublished RCTs that had been likely conducted. Considering that the first aTIV licensure in Italy was for individuals aged ≥12 years [16], other adult-specific data could exist. Second, the RoB of some trials published before 2010 could be inaccurate, as these studies were not drafted according to the available reporting guidelines. Indeed, following development of the consolidated standards of reporting trials, these were not homogeneously adopted by journals, the academic community and clinical trial researchers [81,82]. Here, we adopted a conservative approach and rated these early RCTs at high RoB and therefore our ratings may be biased. Finally, we decided against pooling the continuous variable of relative GMTs of aTIV/aQIV vs.

TIV/QIV, as several studies did not report dispersion measures. We noted that the 95% CIs were more likely reported in case of statistically significant results and by more recently published trials. Omission of studies with no dispersion measures would undoubtedly favor aTIV/aQIV, while the imputation of standard deviations was judged unfeasible owing to a significant number of GMTs without 95% CIs. In any case, our approach is conservative as the overwhelming majority of GMTR point estimates favored aTIV/aQIV.

The major limitation of the current body of evidence is the small sample sizes of most studies. Indeed, one half of the included trials enrolled <100 participants. These studies likely contributed to the substantial heterogeneity in some meta-analyses. The second important limitation is that data on the absolute and relative efficacy and effectiveness of aTIV/aQIV were very limited or lacking altogether. Considering the recent European authorization of aQIV for adults aged 50–64 [15] years, comparative vaccine effectiveness studies in this population group are warranted.

In conclusion, in this study based on 24 publications and 4,628 individuals, we showed that aTIV/aQIV does not raise safety concerns in adults and meets the regulatory criteria for absolute immunogenicity. aTIV/aQIV is generally more immunogenic than non-adjuvanted standard-dose vaccines towards both homologous and heterologous strains. The magnitude of this benefit, however, depends on vaccine component and characteristics of vaccinees, being higher for the A(H3N2) subtype and in adults with immunosuppressive conditions. Very limited data on the relative efficacy and effectiveness of aTIV/aQIV hampers further technology assessment.

## Supporting information

**S1 Fig. Forest plot of absolute seroconversion rates towards vaccine-like A(H1N1) strains 3–4 weeks after one dose of the MF59-adjuvanted seasonal influenza vaccine in non-elderly adults, by immunosuppression status.**
(DOCX)

**S2 Fig. Forest plot of absolute seroconversion rates towards vaccine-like A(H3N2) strains 3–4 weeks after one dose of the MF59-adjuvanted seasonal influenza vaccine in non-elderly adults, by immunosuppression status.**
(DOCX)

**S3 Fig. Forest plot of absolute seroconversion rates towards vaccine-like B strains 3–4 weeks after one dose of the MF59-adjuvanted seasonal influenza vaccine in non-elderly adults, by immunosuppression status.**
(DOCX)

**S4 Fig. Forest plot of absolute seroprotection rates (hemagglutination inhibition titer ≥1:40) towards vaccine-like A(H1N1) strains 3–4 weeks after one dose of the MF59-adjuvanted seasonal influenza vaccine in non-elderly adults, by immunosuppression status.**
(DOCX)

**S5 Fig. Forest plot of absolute seroprotection rates (hemagglutination inhibition titer ≥1:40) towards vaccine-like A(H3N2) strains 3–4 weeks after one dose of the MF59-adjuvanted seasonal influenza vaccine in non-elderly adults, by immunosuppression status.**
(DOCX)

**S6 Fig. Forest plot of absolute seroprotection rates (hemagglutination inhibition titer ≥1:40) towards vaccine-like B strains 3–4 weeks after one dose of the MF59-adjuvanted seasonal influenza vaccine in non-elderly adults, by immunosuppression status.**
(DOCX)

**S7 Fig. Forest plot of difference in seroconversion rates towards vaccine-like A(H1N1) strains 3–4 weeks after one dose of the MF59-adjuvanted or non-adjuvanted seasonal influenza vaccines in non-elderly adults, by immunosuppression status.** (DOCX)

**S8 Fig. Forest plot of difference in seroconversion rates towards vaccine-like A(H3N2) strains 3–4 weeks after one dose of the MF59-adjuvanted or non-adjuvanted seasonal influenza vaccines in non-elderly adults, by immunosuppression status.** (DOCX)

**S9 Fig. Forest plot of difference in seroconversion rates towards vaccine-like B strains 3–4 weeks after one dose of the MF59-adjuvanted or non-adjuvanted seasonal influenza vaccines in non-elderly adults, by immunosuppression status.** (DOCX)

**S10 Fig. Forest plot of difference in seroprotection rates towards vaccine-like A(H1N1) strains 3–4 weeks after one dose of the MF59-adjuvanted or non-adjuvanted seasonal influenza vaccines in non-elderly adults, by immunosuppression status.** (DOCX)

**S11 Fig. Forest plot of difference in seroprotection rates towards vaccine-like A(H3N2) strains 3–4 weeks after one dose of the MF59-adjuvanted or non-adjuvanted seasonal influenza vaccines in non-elderly adults, by immunosuppression status.** (DOCX)

**S12 Fig. Forest plot of difference in seroprotection rates towards vaccine-like B strains 3–4 weeks after one dose of the MF59-adjuvanted or non-adjuvanted seasonal influenza vaccines in non-elderly adults, by immunosuppression status.** (DOCX)

**S13 Fig. Funnel plot of difference in seroconversion rates towards vaccine-like A(H1N1) strains 3–4 weeks after one dose of the MF59-adjuvanted or non-adjuvanted seasonal influenza vaccines in non-elderly adults.** (DOCX)

**S14 Fig. Funnel plot of difference in seroconversion rates towards vaccine-like A(H3N2) strains 3–4 weeks after one dose of the MF59-adjuvanted or non-adjuvanted seasonal influenza vaccines in non-elderly adults.** (DOCX)

**S15 Fig. Funnel plot of difference in seroconversion rates towards vaccine-like B strains 3–4 weeks after one dose of the MF59-adjuvanted or non-adjuvanted seasonal influenza vaccines in non-elderly adults.** (DOCX)

**S16 Fig. Funnel plot of difference in seroprotection rates towards vaccine-like A(H1N1) strains 3–4 weeks after one dose of the MF59-adjuvanted or non-adjuvanted seasonal influenza vaccines in non-elderly adults.** (DOCX)

**S17 Fig. Funnel plot of difference in seroprotection rates towards vaccine-like A(H3N2) strains 3–4 weeks after one dose of the MF59-adjuvanted or non-adjuvanted seasonal**

**influenza vaccines in non-elderly adults.**
(DOCX)

**S18 Fig. Funnel plot of difference in seroprotection rates towards vaccine-like B strains 3–4 weeks after one dose of the MF59-adjuvanted or non-adjuvanted seasonal influenza vaccines in non-elderly adults.**
(DOCX)

**S19 Fig. Frequency of any solicited local/injection site reaction during the first week after one dose of the MF59-adjuvanted seasonal influenza vaccine in non-elderly adults.**
(DOCX)

**S20 Fig. Frequency of solicited injection site pain during the first week after one dose of the MF59-adjuvanted seasonal influenza vaccine in non-elderly adults.**
(DOCX)

**S21 Fig. Frequency of solicited injection site erythema during the first week after one dose of the MF59-adjuvanted seasonal influenza vaccine in non-elderly adults.**
(DOCX)

**S22 Fig. Frequency of solicited injection site induration during the first week after one dose of the MF59-adjuvanted seasonal influenza vaccine in non-elderly adults.**
(DOCX)

**S23 Fig. Frequency of solicited injection site ecchymosis during the first week after one dose of the MF59-adjuvanted seasonal influenza vaccine in non-elderly adults.**
(DOCX)

**S24 Fig. Frequency of any solicited systemic reaction during the first week after one dose of the MF59-adjuvanted seasonal influenza vaccine in non-elderly adults.**
(DOCX)

**S25 Fig. Frequency of solicited fever during the first week after one dose of the MF59-adjuvanted seasonal influenza vaccine in non-elderly adults.**
(DOCX)

**S26 Fig. Frequency of solicited chills during the first week after one dose of the MF59-adjuvanted seasonal influenza vaccine in non-elderly adults.**
(DOCX)

**S27 Fig. Frequency of solicited myalgia during the first week after one dose of the MF59-adjuvanted seasonal influenza vaccine in non-elderly adults.**
(DOCX)

**S28 Fig. Frequency of solicited arthralgia during the first week after one dose of the MF59-adjuvanted seasonal influenza vaccine in non-elderly adults.**
(DOCX)

**S29 Fig. Frequency of solicited headache during the first week after one dose of the MF59-adjuvanted seasonal influenza vaccine in non-elderly adults.**
(DOCX)

**S30 Fig. Frequency of solicited malaise during the first week after one dose of the MF59-adjuvanted seasonal influenza vaccine in non-elderly adults.**
(DOCX)

**S31 Fig. Frequency of solicited nausea during the first week after one dose of the MF59-adjuvanted seasonal influenza vaccine in non-elderly adults.**
(DOCX)

**S32 Fig. Relative risk of any solicited local/injection site reaction during the first week after one dose of the MF59-adjuvanted or non-adjuvanted seasonal influenza vaccines in non-elderly adults.**
(DOCX)

**S33 Fig. Relative risk of solicited injection site pain during the first week after one dose of the MF59-adjuvanted or non-adjuvanted seasonal influenza vaccines in non-elderly adults.**
(DOCX)

**S34 Fig. Relative risk of solicited injection site erythema during the first week after one dose of the MF59-adjuvanted or non-adjuvanted seasonal influenza vaccines in non-elderly adults.**
(DOCX)

**S35 Fig. Relative risk of solicited injection site induration during the first week after one dose of the MF59-adjuvanted or non-adjuvanted seasonal influenza vaccines in non-elderly adults.**
(DOCX)

**S36 Fig. Relative risk of solicited injection site ecchymosis during the first week after one dose of the MF59-adjuvanted or non-adjuvanted seasonal influenza vaccines in non-elderly adults.**
(DOCX)

**S37 Fig. Relative risk of any solicited systemic reaction during the first week after one dose of the MF59-adjuvanted or non-adjuvanted seasonal influenza vaccines in non-elderly adults.**
(DOCX)

**S38 Fig. Relative risk of solicited fever during the first week after one dose of the MF59-adjuvanted or non-adjuvanted seasonal influenza vaccines in non-elderly adults.**
(DOCX)

**S39 Fig. Relative risk of solicited chills during the first week after one dose of the MF59-adjuvanted or non-adjuvanted seasonal influenza vaccines in non-elderly adults.**
(DOCX)

**S40 Fig. Relative risk of solicited myalgia during the first week after one dose of the MF59-adjuvanted or non-adjuvanted seasonal influenza vaccines in non-elderly adults.**
(DOCX)

**S41 Fig. Relative risk of solicited arthralgia during the first week after one dose of the MF59-adjuvanted or non-adjuvanted seasonal influenza vaccines in non-elderly adults.**
(DOCX)

**S42 Fig. Relative risk of solicited headache during the first week after one dose of the MF59-adjuvanted or non-adjuvanted seasonal influenza vaccines in non-elderly adults.**
(DOCX)

**S43 Fig. Relative risk of solicited malaise during the first week after one dose of the MF59-adjuvanted or non-adjuvanted seasonal influenza vaccines in non-elderly adults.**
(DOCX)

**S44 Fig. Relative risk of solicited nausea during the first week after one dose of the MF59-adjuvanted or non-adjuvanted seasonal influenza vaccines in non-elderly adults.**
(DOCX)

**S45 Fig. Frequency of serious adverse events during after one dose of the MF59-adjuvanted seasonal influenza vaccine in non-elderly adults.**
(DOCX)

**S46 Fig. Relative risk of serious adverse events after one dose of the MF59-adjuvanted or non-adjuvanted seasonal influenza vaccines in non-elderly adults.**
(DOCX)

**S1 Table. PRISMA (Preferred Reporting Items for Systematic reviews and Meta-Analyses) checklist.**
(DOCX)

**S2 Table. Algorithm for the research performed on 16 February 2024, by citation database.**
(DOCX)

**S3 Table. List of excluded studies with reasons.**
(DOCX)

**S4 Table. Risk of bias (RoB) assessment of the selected randomized trials.**
(DOCX)

**S5 Table. Risk of bias (RoB) assessment of the selected non-randomized trials.**
(DOCX)

**S6 Table. Extracted data on the comparison of seroconversion and seroprotection rates and geometric mean titer ratios against vaccine-like strains 3–4 weeks after one dose of the MF59-adjuvanted or non-adjuvanted seasonal influenza vaccines in non-elderly adults, by strain, presence of immunosuppressive conditions and serological parameter.**
(DOCX)

**S7 Table. Absolute seroprotection rates towards vaccine-like strains 3–4 weeks after one dose of the MF59-adjuvanted seasonal influenza vaccine in non-elderly adults: A sensitivity analysis by including any hemagglutination inhibition titer threshold (both $\geq$ 1:40 and $\geq$ 1:160).**
(DOCX)

**S8 Table. Meta-analysis of absolute and relative seroconversion and seroprotection rates towards vaccine-like strains 3–4 weeks after one dose of the MF59-adjuvanted or non-adjuvanted seasonal influenza vaccines: A sensitivity analysis by excluding studies with some elderly participants.**
(DOCX)

**S9 Table. Univariable meta-regression analysis to investigate sources of heterogeneity in relative seroconversion and seroprotection rates towards vaccine-like strains 3–4 weeks after one dose of the MF59-adjuvanted or non-adjuvanted seasonal influenza vaccines.**
(DOCX)

**S10 Table. Extracted data on the comparison of seroconversion and seroprotection rates and geometric mean titer ratios against heterologous strains 3–4 weeks after one dose of the MF59-adjuvanted or non-adjuvanted seasonal influenza vaccines in non-elderly adults,**

by strain, presence of immunosuppressive conditions and serological parameter.
(DOCX)

**S11 Table. Meta-analysis of relative seroconversion and seroprotection rates towards heterologous strains 3–4 weeks after one dose of the MF59-adjuvanted or non-adjuvanted seasonal influenza vaccines in non-elderly adults, by heterologous strain.**
(DOCX)

**S12 Table. Extracted data on the comparison of seroprotection rates and geometric mean titer ratios towards vaccine-like strains 3–9 months after one dose of the MF59-adjuvanted or non-adjuvanted seasonal influenza vaccines in non-elderly adults, by strain, immunosuppression status and time post-vaccination.**
(DOCX)

**S13 Table. Meta-analysis of absolute and relative seroprotection rates towards vaccine-like strains six months after one dose of the MF59-adjuvanted or non-adjuvanted seasonal influenza vaccines in non-elderly adults, by strain and time post-vaccination.**
(DOCX)

**S14 Table. Extracted data on the comparison of local/injection site and systemic solicited adverse reactions during the first week after one dose of the MF59-adjuvanted or non-adjuvanted seasonal influenza vaccines in non-elderly adults, by adverse reaction.**
(DOCX)

## Author Contributions

**Conceptualization:** Alexander Domnich, Elettra Fallani, Marco Salvatore.

**Data curation:** Alexander Domnich, Carlo-Simone Trombetta.

**Formal analysis:** Alexander Domnich, Carlo-Simone Trombetta.

**Funding acquisition:** Marco Salvatore.

**Investigation:** Alexander Domnich, Carlo-Simone Trombetta.

**Methodology:** Alexander Domnich.

**Project administration:** Elettra Fallani.

**Resources:** Marco Salvatore.

**Supervision:** Marco Salvatore.

**Validation:** Alexander Domnich, Carlo-Simone Trombetta.

**Writing – original draft:** Alexander Domnich.

**Writing – review & editing:** Carlo-Simone Trombetta, Elettra Fallani, Marco Salvatore.

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
