## [Decision Letter · Decision Letter 0]

4 Sep 2024

Immunogenicity and safety of the MF59-adjuvanted seasonal influenza vaccine in non-elderly adults: A systematic review and meta-analysis

PONE-D-24-18646

Dear Dr. Salvatore,

We’re pleased to inform you that your manuscript has been judged scientifically suitable for publication and will be formally accepted for publication once it meets all outstanding technical requirements.

Kind regards,

Mrinmoy Sanyal, PhD

Academic Editor

PLOS ONE

 [This work was supported by Seqirus S.r.L. Seqirus S.r.L. was involved in study design, decision to publish and preparation of the manuscript.].    

Please respond by return e-mail so that we can amend your financial disclosure and competing interests on your behalf.

[EF and MS are employees Seqirus S.r.L. AD was previously an employee of Seqirus S.r.L. AD and CST received a fee from Seqirus S.r.L. for conducting this systematic review.]. 

Please respond by return email with your amended Competing Interests Statement and we will change the online submission form on your behalf.

Reviewers' comments:

Reviewer's Responses to Questions

**Comments to the Author**

1. Is the manuscript technically sound, and do the data support the conclusions?

Reviewer #1: Yes

Reviewer #2: Yes

2. Has the statistical analysis been performed appropriately and rigorously? 

Reviewer #1: Yes

Reviewer #2: Yes

3. Have the authors made all data underlying the findings in their manuscript fully available?

Reviewer #1: Yes

Reviewer #2: Yes

4. Is the manuscript presented in an intelligible fashion and written in standard English?

Reviewer #1: Yes

Reviewer #2: Yes

5. Review Comments to the Author

Reviewer #1: The research article of Marco Salvatore et al. ‘Immunogenicity and safety of the MF59-adjuvanted seasonal influenza vaccine in nonelderly adults: A systematic review and meta-analysis.’ represents a systematic literature review, conducted according to the current guidelines and based on solid statistical methodology and comprehensive search criteria. The authors identified and carefully reviewed 24 publications assessing immunogenicity and safety of aTIV/aQIV in adults 18 to 64 years, conducted since aTIV licensure in 1997. Most publications evaluated the vaccine's immunogenicity using hemagglutinin inhibition assay, with very few studies assessing cell-mediated immunity, antibody persistence, or immune response against heterologous influenza strains.

This work of specific interest based on recent extension of aQIV indication in EU from 65 years and above to ≥50 years, and associated questions on whether the administration of the vaccine to younger adults has potential benefits.

Overall, the authors demonstrated that the magnitude of incremental immunological benefits of MF59-adjuvanted vaccine over non-adjuvanted conventional influenza vaccines is relatively small (7.1% to 8.5% for SRR difference and 3.3% to 5.3% for SPR difference). The differences are more prominent in the immunocompromised populations.

The key limitation of the study, as correctly highlighted by the authors, is the absence of reliable efficacy and effectiveness data. As such, assessing whether observed immunological benefits are translated into better clinical protection against influenza disease is very difficult.

The review is comprehensive, and the key outcomes are consistent with previously published systematic reviews. The manuscript also highlights the overall deficiency of available clinical data and the inability to answer whether MF59 adjuvanted vaccines extend the breadth of neutralizing antibodies and associated protection.

The publication might interest infectious disease specialists, vaccinologists, and public health experts.

It would be great to have an author's position regarding the overall benefit/risk profile of MF59-adjuvanted influenza vaccines in young adults based on the totality of available data and the outcome of the presented meta-analysis.

Reviewer #2: MF59-adjuvanted trivalent influenza vaccine (aTIV) has been approved in Italy for use in individuals aged 12 years and older. This approval allows for the vaccination of young adults, particularly benefiting those who may have a weaker immune response to standard vaccines. It is important to refer to specific data for the most accurate and up-to-date recommendations regarding vaccine use.

In this research authors systematically appraise data on the immunogenicity, efficacy and safety of aTIV/aQIV in non-elderly adults.

The review and meta-analysis of adjuvants for influenza vaccines faced all key criteria to ensure the research is comprehensive, rigorous and reliable.

- Reporting Standards: Followed established reporting guidelines PRISMA (Preferred Reporting Items for Systematic Reviews and Meta-Analyses), to ensure transparency and completeness in reporting the review process and findings

- Population: working-age adults <65 years

- Intervention: a single dose of the MF59-adjuvanted formulations aTIV or aQIV (A(H1N1), A(H3N2), and B for trivalent formulations, and A(H1N1), A(H3N2), and B/Victoria and B/Yamagata for quadrivalent formulations)

- Comparison: standard influenza vaccine without adjuvants

- Outcomes: antibody titers measured in the hemagglutination-inhibition (HAI) assay

- Search Strategy: used databases MEDLINE (via Ovid), Biological Abstracts (via Ovid), Web of Science and Cochrane Library, Google Scholar.

- Data Extraction: (i) citation record; (ii) study location; (iii) influenza season; (iv) study design; (v) study population and main population characteristics (age and co-morbidities); (vi) sample size per study arm; (vii) outcome domains evaluated in the study; (viii) vaccine and test strains used; (ix) point estimates for the outcomes of interest with any available dispersion measures; (x) funding source; (xii) other potentially relevant information.

- Quality Assessment: Assess the quality and risk of bias of the included studies used Cochrane RoB 2 tool and ROBINS-I tool.

- Statistical Analysis: both fixed-effects (FE) and random-effects (RE) models were applied

-Sensitivity Analysis: The first concerned the operational definitions of SPR. The second, they excluded studies with overlapping study populations. The third, procedure was implemented to estimate the number of potentially unavailable studies and to adjust pooled estimates for publication bias

The manuscript is generally well-written. The authors were accurate and critical in their conclusions and pointed out the study's limitations.

6. PLOS authors have the option to publish the peer review history of their article (what does this mean?). If published, this will include your full peer review and any attached files.

Reviewer #1: No

Reviewer #2: No

---

## [Editor Report · Acceptance letter]

11 Oct 2024

PONE-D-24-18646 

PLOS ONE

Dear Dr. Salvatore, 

I'm pleased to inform you that your manuscript has been deemed suitable for publication in PLOS ONE. Congratulations! Your manuscript is now being handed over to our production team.

Kind regards, 

on behalf of

Dr. Mrinmoy Sanyal 

Academic Editor

PLOS ONE